# A Universal Model for Ultrasonic Energy Transmission in Various Media

**DOI:** 10.3390/s24196230

**Published:** 2024-09-26

**Authors:** Yufei Ma, Yunan Jiang, Chong Li

**Affiliations:** James Watt School of Engineering, University of Glasgow, Glasgow G12 8QQ, UK; y.ma.1@research.gla.ac.uk (Y.M.); 2602587j@student.gla.ac.uk (Y.J.)

**Keywords:** ultrasonic energy transfer (UET), ultrasonic link, path loss, empirical model

## Abstract

This study presents a comprehensive model for ultrasonic energy transfer (UET) using a 33-mode piezoelectric transducer to advance wireless sensor powering in challenging environments. One of the advantages of UET is that it is not stoppable by electromagnetic shielding and can penetrate metal. Existing models focus on feasibility and numerical analysis but lack an effective link between input and output power in different media applications. The proposed model fills this gap by incorporating key factors of link loss, including resonant frequency, impedance matching, acoustic coupling, and boundary conditions, to predict energy transfer efficiency more accurately. The model is validated through numerical simulations and experimental tests in air, metal, and underwater environments. An error analysis has shown that the maximum error between theoretical and experimental responses is 3.11% (air), 27.37% (water), and 1.76% (aluminum). This research provides valuable insights into UET dynamics and offers practical guidelines for developing efficient wireless powering solutions for sensors in difficult-to-access or electromagnetically shielded conditions.

## 1. Introduction

Over the past decade, ultrasonic energy transmission (UET) has attracted extensive interest in wireless power transfer [1,2,3]. Potential applications include powering sensors sealed in metallic boxes, implantable devices, and underwater scenarios [4,5,6,7,8]. Notably, UET systems of equivalent dimensions offer superior transmission distances compared to capacitive and inductive coupling [1,9]. This capability has unlocked the potential for powering distributed sensors, addressing size limitations, and promoting miniaturization [10]. Additionally, UET relies on mechanical waves, which avoid electromagnetic constraints and safely penetrate both metallic barriers and bodily tissues [11]. Consequently, UET’s application in medical implants and metallic sealing has garnered significant attention [12,13,14].

One of the key factors in UET applications is understanding how ultrasound propagates between the transmitter and receivers, as transmission efficiency significantly affects system design. Wilt et al. [15] proposed a one-dimensional pressure transfer model to establish a piezoelectric-based ultrasonic acoustic power channel model. This model constructs a simple channel for energy transmission and verifies the excitation response. However, it only considers the electrical domain, with insufficient discussion of the mechanical domain, thus not fully demonstrating the energy transfer process. Du et al. [16] developed a two-dimensional energy transfer model based on the radial and axial directions of piezoelectric transducers, improving the accuracy of the equivalent circuit of traditional piezoelectric transducers. Nonetheless, this model focuses on the performance of ring-shaped piezoelectric transducers, whereas most applications utilize circular plate transducers and do not demonstrate performance at various stages of energy transfer. Mendonca et al. [17] demonstrated a modeling method for multiphysics networks in acoustic systems, including the transmission medium, distance, and frequency range. Their model incorporates physical components in both the acoustic and mechanical domains but only considers changes in resonant frequency without addressing the resulting changes caused by variations in other physical components across different applications.

Jiao et al. [18] established a semi-analytical model of ultrasonic power transfer, discussing the accuracy of the model and the improvement in parameters. This model accounts for transducer and transmission distances but only verifies performance in a metallic medium. As the medium changes, the impedance of the transducer also changes, limiting the model’s applicability across different media. Wu et al. [19] proposed and validated new equivalent circuit models of piezoelectric and non-piezoelectric components considering UET channel losses in underwater power transfer (UWPT) systems. They used two-port T parameters to describe the UET channel, calculating input impedance, input power, and output power with the proposed model. However, the parameters for each link loss model are difficult to obtain or measure, and the model cannot show the losses in the system at each stage.

In response to the complexity and limitations of existing UET models, we propose a straightforward transmission model based on extensive experimentation. The contribution of this paper can be summarized as follows: The ultrasonic energy transmission model, combined with the actual application environment, is established to illustrate the entire power transfer process from the transmitting source to the load, reflecting the impact of parameter changes in the actual application of the transducer on the system.The model thoroughly analyzes the link losses in ultrasonic energy transmission, developing and verifying results across different application scenarios, including the influence of transducer size, frequency, and transmission medium.The model provides essential guidance for designing ultrasonic energy transmission systems. By considering link losses and the application environment, methods for enhancing system efficiency can be identified and optimized.

The paper is organized as follows. Section 1 introduces the background and motivation of the work. Section 2 elaborates on the UET model. Section 3 details the experimental tests, and Section 4 presents the numerical calculations alongside test comparisons, followed by discussions. Finally, our conclusions are summarized in Section 5.

## 2. The UET Model

### 2.1. System Architecture

The UET model provides an abstract representation of the ultrasonic energy transmission process, analyzing and predicting the energy transmission relationships within the system. It primarily encompasses the system architecture and transmission link losses. The UET model presented in this paper illustrates the conditions for achieving maximum power transfer to the load, e.g., a sensor. As depicted in Figure 1, a UET system consists of a pair of transmitters (Tx) and receivers (Rx). The Tx is powered by a signal source (P_1_) through an impedance-matching network. The energy, in an ultrasonic form, propagates through a medium before being sensed by the Rx. Between Rx and the load (P_2_), another impedance-matching network is often used.

The energy loss between P_1_ and P_2_ constitutes the transmission link loss. Therefore, the system transmission model can be expressed as follows:(1)$$P_{2}\;\left(dBm\right)=P_{1}\left(\mathrm{dBm}\right)-\mathrm{Transmission}\;\mathrm{Link}\;\mathrm{Loss}\;\left(\mathrm{dB}\right).$$

Transmission link loss will be comprehensively discussed in relation to various application environments. These factors include transducers of different sizes and resonant frequencies, transmission media with varying densities, impedance performance across different media, varying transmission distances, and the constructive and destructive interference of waves. Notably, in this model, the mechanical vibration of the piezoelectric transducer propagates along the axial direction, corresponding to mode-33 (z-axis).

### 2.2. Transmission Link Losses

The primary losses in a UET system stem from electrical reflection, acoustic reflection, transducer loss, and medium transmission loss [3,20]. As the resonant frequency increases, both electrical impedance matching and acoustic matching in the system become increasingly complex [21]. The losses observed in this study are illustrated in Figure 2. The individual components of loss are discussed below in terms of energy transfer pathways.

#### 2.2.1. Electrical Loss

Electrical loss refers to the return loss due to reflection when energy (in the form of an electrical signal) reaches the transducer, which usually occurs at impedance mismatches or discontinuities [22]. Electrical loss is measured relatively and expressed by:(2)$$\mathrm{Electrical}\;\mathrm{Loss}\;\left(\mathrm{dB}\right)=-10\log\left(1-{\left|\Gamma_{E}\right|}^{2}\right)$$
where $\Gamma_{E}$ is the reflection coefficient, and given by:(3)$$\Gamma_{E}=\frac{Z_{E1}-Z_{E2}}{Z_{E1}+Z_{E2}},$$
where *Z*_E1_ is the electrical impedance of the driver, which can be the impedance of the signal generator or the transducer receiver. Z_E2_ is the electrical impedance of the load, which can be the impedance of the load or the transducer transmitter. Electrical losses can be determined by the impedance values of the transducer, signal generator, and load.

#### 2.2.2. Transducer Loss

Losses in piezoelectric transducers mainly include dielectric loss, elastic loss, and piezoelectric loss [23]. These losses are characterized by dielectric permittivity ($\beta_{33}^{S}$), elastic stiffness constant ($C_{33}^{D}$), and piezoelectric stiffness constant ($h_{33}$). Accurately measuring these coefficients in practical applications is challenging. Thus, a simpler method is used to express transducer loss.

In this study, energy is transferred through the medium as mechanical energy. Therefore, the electromechanical coupling coefficient $k^{2}$ is a key metric for evaluating transducer losses [24]. The $k^{2}$ is closely related to the efficiency of energy conversion between electrical and mechanical forms. Specifically, $k^{2}$ represents the ratio of stored mechanical energy to input electrical energy, indicating conversion efficiency. It also applies to the inverse scenario, representing the ratio of stored electrical energy to input mechanical energy [23].
(4)$$k^{2}=\frac{\mathrm{mechanical}\;\mathrm{energy}\;\mathrm{converted}\;\mathrm{into}\;\mathrm{electrical}\;\mathrm{energy}}{\mathrm{input}\;\mathrm{mechanical}\;\mathrm{energy}},$$
or
(5)$$k^{2}=\frac{\mathrm{electrical}\;\mathrm{energy}\;\mathrm{converted}\;\mathrm{into}\;\mathrm{mechanical}\;\mathrm{energy}}{\mathrm{input}\;\mathrm{electrical}\;\mathrm{energy}},$$
where $k^{2}$ is related to the vibration mode and shape of the transducer. When the electric field is parallel to the direction of polarization, the induced strain is in the same direction as the electric field. For discs and rods shaped transducers, $k^{2}$ can be calculated as:(6)$$k_{d}^{2}=\frac{\pi }{2}\frac{f_{n}}{f_{m}}\tan\left[\frac{\pi }{2}\left(\frac{f_{n}-f_{m}}{f_{m}}\right)\right],$$

For transducers with other shapes, a general model is expressed as:(7)$$k_{eff}^{2}=\frac{\left(f_{n}^{2}-f_{m}^{2}\right)}{f_{n}^{2}},$$
where *f_m_* is minimum impedance frequency (resonance frequency), and *f_n_* is maximum impedance frequency (anti-resonance frequency). However, $k^{2}$ does not account for dielectric losses or mechanical losses, nor recovery of unconverted energy. Therefore, the transducer loss is modified as follows:(8)$$\mathrm{Transducer}\;\mathrm{loss}\;\left(\mathrm{dB}\right)=-10\log(k^{2}),$$

#### 2.2.3. Acoustic Loss

Acoustic loss primarily occurs at the boundaries where the transducer interfaces with the transmission medium [25]. The acoustic impedance of the transducer typically differs from that of the transmission medium, resulting in boundary reflections.

Similar to electrical losses, the expression for acoustic boundary loss is governed by:(9)$$\mathrm{Acoustic}\;\mathrm{Loss}\;\left(\mathrm{dB}\right)=-10\log\left(1-{\left|\Gamma_{A}\right|}^{2}\right),$$
where $\Gamma_{A}$ is the reflection coefficient, defined as: (10)$$\Gamma_{A}=\frac{Z_{A2}-Z_{A1}}{Z_{A2}+Z_{A1}},$$
where *Z_A1_* and *Z_A2_* are the acoustic impedances of the transducer and the transmission medium.

#### 2.2.4. Medium Loss

Medium loss refers to the energy loss that occurs as the transducer’s emitted energy transfers through the propagation medium, influenced by the medium’s density and rigidity [26]. To investigate the medium loss, it is first necessary to establish a model of the transducer emitting ultrasonic waves in various media.

Transducers play a significant role in influencing the ultrasonic energy transmission system. Although some researchers model the transducer emission as a point source, in practical applications, the transducer’s radius is comparable to the ultrasonic wavelength and cannot be ignored. Therefore, the transducer is modeled as an infinite planar piston to more accurately reflect its practical application. When the transducer has a fixed back surface, it is modeled as an infinite plane, as shown in Figure 3.

This study primarily explores energy transfer from a transmit transducer to a receive transducer over varying transmission distances within different media. Using the infinite plane piston acoustic model, we developed a model to quantify medium loss by analyzing energy transmission from the transducer’s transmitter to the receiver. Although the classical circular piston acoustic model in an infinite rigid baffle has been extensively studied by researchers such as David and Bies [27], Kinesler et al. [28], Pierce [29], Meyer and Neumann [30], we propose an improved model for analyzing and calculating ultrasonic energy transmission loss in different media. This model utilizes the reciprocal two-port network characteristics of the transducer, with pressure as an intermediate variable.

As illustrated in Figure 4, when the transmitter input power is *P_1_*, the pressure at observation point *O*, located a distance *r* away, is *p*. According to two-port reciprocity, if the pressure at observation point *O* is *p*, then the transducer’s output power is *P_1_*.

Replacing observation point *O* with the receiver and setting the transmission distance *r* to zero yields the received power *P_2_*.

Based on the infinite planar piston model, the total pressure *p* at observation point *O* is obtained by integrating over the surface of the piston as follows:(11)$$p=\frac{j\rho ck}{2\pi r}Ue^{j\left(\omega t-kr\right)}\int_{0}^{a}\sigma \;d\sigma \int_{0}^{2\pi }e^{jk\sigma \sin\theta \cos\psi }\;d\psi ,$$
where ρ is the medium density, c is the speed of sound in the medium, U is the velocity amplitude of piston surface, k $=\frac{2\pi f}{c}$, ω is the angular velocity, a is the piston radius, and r is the transmission distance.

The piston exhibits different behaviors in the near field and far field. In the far field, the piston shows significant directionality. In contrast, the near-field pressure field contains local maxima and minima, and the directionality is less pronounced. Therefore, the far-field and near-field conditions will be discussed separately. Since this study focuses on the relationship between energy transmission and transmission distance, the observation point is placed on the z-axis at a distance *r*. Consequently, the relationship between pressure and transmission distance is as follows:

For far field,
(12)$$p=j\rho c\frac{Uka^{2}}{2r}e^{j\left[\omega t-\frac{k}{2}\left(\sqrt{r^{2}+a^{2}}+r\right)\right]},$$

For near field,
(13)$$p=j2\rho cUe^{j\omega t}\sin\left[\frac{k}{2}\left(\sqrt{r^{2}+a^{2}}-r\right)\right]e^{-j\frac{k}{2}\left(\sqrt{r^{2}+a^{2}}+r\right)},$$
where the condition for a point of observation in the far field is $r\gg \frac{fa^{2}}{c}$.

To simplify testing and reduce system parameter complexity, the piezoelectric coefficient (*d*_33_) of the transducer is introduced. The vibration speed of the medium is equated to that of the transducer, giving the expression for U:(14)$$U=\frac{ds}{dt}=\frac{d(d_{33}\times V)}{dt}.$$

In the UET system, the transducer’s driving voltage is expressed as Vsin(ωt), and thus, the U is given by:(15)$$U=d_{33}\times V\times \omega ,$$

Therefore, the transducer power is expressed as:(16)$$P=\frac{V^{2}}{Z_{E}}=\frac{1}{Z_{E}}{\left(\frac{U}{d_{33}\times \omega }\right)}^{2}.$$

Due to transmitter-receiver reciprocity, the relationship between surface vibration velocities at distance *r*, based on the transmission model, is expressed as:

For far field,
(17)$$\frac{U_{1}}{U_{2}}=\frac{4r}{ka^{2}}\sin\frac{ka}{2}e^{j\left[\frac{k}{2}\left(\sqrt{r^{2}+a^{2}}+r-a\right)\right]},$$

For near field,
(18)$$\frac{U_{1}}{U_{2}}=\sin\frac{ka}{2}\csc\left[\frac{k}{2}\left(\sqrt{r^{2}+a^{2}}-r\right)\right]e^{j\left[\frac{k}{2}\left(\sqrt{r^{2}+a^{2}}+r-a\right)\right]}.$$

The medium loss is given by:(19)$$\mathrm{Medium}\;\mathrm{loss}\;\left(\mathrm{dB}\right)=-10\log\left[\frac{Z_{E2}}{Z_{E1}}\times {\left(\frac{U_{1}}{U_{2}}\right)}^{2}\right].$$

The energy at the receiver is influenced by both transmission losses and interference. Transmission interference arises from the superposition of ultrasound waves (mainly longitudinal waves) between the transmitter and receiver within the medium. Constructive and destructive interference occurs as the transmission distance varies. Therefore, the pressure at the receiver is minimum at nodes and maximum at antinodes. 

In free space, the positions of nodes and antinodes based on transmission distance and wavelength are:

Nodes
(20)$$r=2n\times \frac{1}{4}\lambda ,$$

Antinodes
(21)$$r=\left(2n+1\right)\times \frac{1}{4}\lambda ,$$
where *n* is the number of the nodes and antinode, and *λ* is the wavelength.

In enclosed spaces, transmission interference arises from the receiver surface and boundaries, leading to more complex results at the receiver. In UET systems, the receiver should be positioned at an antinode to maximize received energy.

In summary, according to Equation (1), the energy transfer of the UET model is:(22)$$P_{2}\;\left(dBm\right)=P_{1}\left(\mathrm{dBm}\right)-\mathrm{electrical}\;\mathrm{Loss}\left(\mathrm{dB}\right)-\mathrm{transducer}\;\mathrm{Loss}\left(\mathrm{dB}\right)-\;\mathrm{acoustic}\;\mathrm{Loss}\left(\mathrm{dB}\right)-\mathrm{medium}\;\mathrm{loss}\left(\mathrm{dB}\right)-\alpha .$$
where *α* is a correction factor introduced to compensate for the discrepancy between the actual measured loss and the theoretically calculated loss, which can be determined through experimental measurements. 

The proposed UET model quantifies the ultrasonic energy transfer process by analyzing link losses, refining the energy flow, and assisting in predicting the output energy based on the input. By examining the link losses, the corresponding loss ratio can be determined. Adjusting UET model parameters, including transducer radius, frequency, and medium density, helps identify optimal solutions to minimize losses and optimize the system.

## 3. Materials and Methods

The transmission model of ultrasonic energy transfer was benchmarked by investigating the effects of transducers with different frequencies and sizes on energy transmission. The transmission media tested includes three materials with significant density differences: air, water, and 6082 aluminum alloy. The test setup is illustrated in Figure 5.

### 3.1. Link Loss Measurement

#### 3.1.1. For Electrical Loss

Electrical losses arise from impedance mismatches between the driving source and the ultrasonic transducer, based on the electrical reflection model. The test procedure includes measuring the impedance of both the transducer in various application environments and the driver circuit. The driving source, consisting of a signal generator and power amplifier, is set with a standard impedance of 50 ohms. Notably, the impedance of the transducer varies with changes in the application environment. Therefore, the first step is to measure the transducer’s impedance under different environmental conditions. The impedance responses of two ultrasonic transducers were tested in three transmission media. Impedance is mainly influenced by the pressure exerted on the transducer’s surface by the transmission medium: in air and water, this pressure varies with height, while in the metallic medium, it is affected by the force exerted on both ends of the transducer.

Given that changes in atmospheric pressure are minimal and difficult to measure in the laboratory, the experiment focused on the impedance variation in water and aluminum mediums. The pressure on the transducer’s surface was altered by adjusting the depth of the transducer in water and by adjusting the force applied in the aluminum medium.

#### 3.1.2. For Transducer Loss

Using the transducer impedance test results, the resonant frequency (*f_m_*) and anti-resonant frequency (*f_n_*) were identified, and the electromechanical coupling coefficient ($k_{d}^{2}$). was calculated using Equation (8). To measure transducer losses, the transmitting and receiving ends were directly connected to create a medium-free energy transmission channel. The correction factors for transducer losses were determined by comparing the transmitted and received energy while accounting for electrical losses.

#### 3.1.3. For Acoustic Loss

Acoustic loss, primarily boundary loss. It can be calculated using Equation (9) according to the material parameters, including density, acoustic velocity, and acoustic impedance. Since the transducer’s packaging is made of aluminum, its acoustic parameters match those of aluminum.

#### 3.1.4. For Medium Loss

Medium loss was measured by varying the transmission distance, following the transmission models in Equations (17) and (18), or by analyzing the relationship between transmitted and received power. As medium loss is influenced by constructive and destructive interference, this test was combined with transmission distance analysis. The test distances and step sizes were set according to the wavelength of ultrasonic waves in different media.

### 3.2. Experimental Setup

To verify the UET model and transmission model, we conducted experimental validations. The test system and setups are illustrated in Figure 6.

An AFG-21005 (Northants, UK) function generator and a WA301 (Cambridge, UK) power amplifier were used to drive the transducer transmitter. The amplifier had a maximum output peak-to-peak voltage of 30 V and an output impedance of 50 Ω. The output voltage of the transducer in different media was monitored using an HP-54600B oscilloscope. The resonant frequency and impedance of the transducer in various media were measured using an Agilent 4294A (Santa Rosa, CA, USA) impedance analyzer. Additionally, an HP-E4404B (Santa Rosa, CA, USA) spectrum analyzer, with a 50 Ω input impedance, was employed to measure the output power from the receiving transducer.

Table 1 shows the parameters required for the test system [31].

Two types of ultrasonic transducers were used for testing: transducer A is MCUSD18A40S09RS-30C from Multicomp (Leeds, UK), and transducer B is 328ET/R250 from Pro-Wave Electronic Co., Ltd. (Kaohsiung, Taiwan).

Transducer A was used both as a transmitter and receiver (T/R), while transducer B operated as an independent transmitter (T) and receiver (R). The transducer is rigidly mounted to the test platform via a 3D-printed bracket to ensure compliance with the infinite plane piston model for the transducer. A linear translation stage (LTS) rail from Thorlabs, Inc. (Newton, NJ, USA) was used to vary the distance between the transmitting and receiving transducers.

The transmission media examined in this study included air, water, and 6082 aluminum alloy. For air, the testing distance ranged from 0 mm to 290 mm with a step size of 0.1 mm. In water, the same distance range was tested but with a step size of 1 mm. For aluminum, due to the challenges associated with modifying the metal’s thickness and the actual requirements of metal penetration applications, the tests were conducted at fixed distances of 5 mm, 10 mm, 15 mm, 20 mm, 25 mm, and 30 mm.

## 4. Results and Discussion

This section presents the results of the comparison of model predictions with experimental observations. The UET models were validated in different media and with two piezoelectric transducers (A and B).

**I.** Impedance variations across medium

While the transducer’s impedance remains stable in the air, its characteristics demonstrate significant changes when used in different environments. Figure 7 illustrates the impedance characteristics of transducer A(T/R) and transducer B(T) and B(R) in various media.

The resonant frequency and impedance of the transducer change based on the environment, influenced by the pressure on the transducer surface. For air, this pressure is governed by atmospheric conditions, while for water, it depends on immersion depth, and for metal, it is influenced by the applied force. The result for the impedance of transducer significant variations was observed in water and aluminum.

For air, Figure 7a,b shows the performance of the transducer under laboratory conditions. In practical applications, the condition that affects atmospheric pressure is altitude. Since altitude cannot be significantly changed in the laboratory, only the test results under laboratory conditions are shown.

For water, Figure 7c,e shows the performance of the transducer underwater. Increasing depth decreases the transducer’s resonant frequency, increases its minimum impedance, and decreases its maximum impedance. Environmental changes can cause frequency and impedance mismatches, which in turn impact both electrical and transducer losses in the link.

For aluminum, Figure 7d,f shows the performance of the transducer attached to aluminum. Increasing the force on the transducer significantly raises its resonant frequency, slightly increases the minimum impedance, and markedly reduces the maximum impedance. Similarly, changes in the application environment alter the transducer’s resonant frequency and impedance, leading to variations in both electrical and transducer losses in the link.

Overall, for transducers operating under different environmental conditions, calculating the losses individually based on the model improves its accuracy compared to models that do not account for link loss. In addition, the results also show that in order to ensure the matching of the frequency and impedance of the system, it is necessary to adjust the frequency and impedance matching network of the driving source according to the resonant frequency and impedance of the actual application, rather than relying on a driving source with a fixed frequency and fixed impedance matching network.

**II.** Loss calculation

Table 2 presents the electrical, transducer, and acoustic losses for both transducers under specific application conditions (in water depth = 150 mm, in aluminum force = 5 N) based on the system model and impedance data.

In a fixed application environment, the system’s three losses remain constant. Electrical loss is primarily determined by the impedance of both the signal source and the transducer. When the application environment changes, variations in electrical loss occur due to changes in the transducer’s impedance. For example, the transducer’s impedance experiences greater fluctuations underwater than when applied to aluminum. As a result, the electrical loss underwater is more pronounced than that observed with aluminum, especially when compared to the loss in air. While the inherent transducer loss of the transducer is governed by its performance characteristics, environmental factors still influence the magnitude of these losses. For instance, the transducer’s loss differs between air and water, though the variation is relatively small.

Conversely, acoustic losses differ significantly across transmission media such as air, water, and aluminum, primarily due to their vastly different acoustic impedances. Acoustic impedance increases with the density of the medium. Although coupling agents are often used in ultrasonic inspection to match acoustic impedance, they also dissipate energy as heat in ultrasonic energy transmission (UET) systems. Furthermore, surface imperfections between the transducer and metal can cause the formation of air gaps, leading to actual acoustic losses that exceed theoretical predictions.

**III.** Medium losses and interference

Medium loss results from material properties, and constructive and destructive interferences occur based on transmission distance, as described by Equation (19). Figure 8 shows the pressure amplitude versus distance in an ideal, reflection-free space.

In an ideal free space without reflection or interference, energy decreases nonlinearly as transmission distance increases. This behavior is influenced by the material properties and ultrasonic frequency.

In practical applications, reflections are inevitable, even in an anechoic chamber. These reflections introduce interference, as illustrated in Figure 9, which shows the predicted and measured results of received power across three different media at various distances.

As the transmission distance increases, the energy received by the transducer decreases, aligning with the model. However, interference causes energy fluctuations with peaks and troughs.

In the air, reflections from the receiver’s surface generate interference at 1/4 wavelength intervals. The predicted values from the model align well with the experimental results. However, for transducer A, when the transmission distance is less than 29 mm, the transmission operates in the near field, resulting in a larger discrepancy between the predicted and measured values. At transmission distances exceeding 200 mm, the measured values fall below the predicted values, primarily due to the reduced energy received. Similarly, for transducer B, when the transmission distance is less than 47 mm, the near-field effect causes a significant error between the test and predicted values. For distances greater than 250 mm, the measured values are lower than the predicted values, again attributed to low received energy.In water, when the transmission distance is less than approximately 16 mm, the system operates in the near field. In water, the ultrasonic wavelength of transducer A is 61.8 mm, and that of transducer B is 82.4 mm. However, the test water tank is limited in size, measuring only 370 × 160 × 220 mm, resulting in significant energy reflections from the tank walls. These reflections lead to multiple wave interferences, causing irregular peaks and troughs. As the receiver approaches the water tank wall, the reflected energy increases, introducing significant errors in the test results, particularly when the transmission distance exceeds 200 mm. The discrepancies between the test and predicted values are primarily constrained by the limited size of the water tank, but the overall trend is in line with expectations.In aluminum alloys, the model employs a correction factor to correct errors resulting from metal surface roughness and air gaps. Moreover, due to the short transmission distance in the metal medium and the high energy levels at the receiver, the difference between the measured and predicted values is minimal.

The model’s predicted trends align with the measured trends across all three media. In order to quantitatively evaluate the models, an error analysis is performed. The error analysis technique is a mean relative average error. The average error can be calculated as:(23)$$\bar{e}\;=\sqrt{{\left(\frac{1}{N}\overset{N}{\underset{i=1}{\sum }}\left(\frac{X_{Ei}-X_{Si}}{X_{Si}}\right)\right)}^{2}}\times 100,$$
in which, $X_{Ei}$ is the experimental value, and $X_{Si}$ is predicted simulation value. N is the number of elements. It is worth noting that the calculated average error value is decided by the effective test range for different media. For air, transducer A’s calculated range is from 29 mm to 200 mm, and transducer B’s calculated range is from 47 mm to 250 mm. For water, the calculated range is from 21 mm to 200 mm, and transducer B’s calculated range is from 20 mm to 200 mm. For aluminum, only six points were calculated.

Error analysis was performed for all the results presented in this section. The purpose is to quantitatively validate the models by calculating the percentage error between experimental and theoretical results, providing a clear metric for evaluating the proposed method. The summarized error analysis is shown in Table 3.

For both air and aluminum, the average error remains below 3.2%. However, in the water, the error is higher, primarily due to constraints in the testing environment. The water tank used for the experiment was insufficiently large, resulting in multiple reflections of the ultrasonic waves off the tank walls, which interfered with the results. Utilizing a larger tank would reduce wall interference, thus decreasing the model’s error.

**IV.** Optimal energy transfer and loss mitigation

In energy transfer systems, maximizing energy transfer is critical, and the transducer should ideally be positioned at a peak point. In metal environments, distance constraints often prevent adjustment, but frequency tuning can align with peak points since the peak is wavelength dependent.

This analysis of link losses highlights strategies for improving system efficiency:Electrical losses can be minimized by designing an impedance-matching network or choosing components with appropriate impedance [32].Transducer losses can be reduced by utilizing high $k^{2}$ and wide-bandwidth transducers, as they can accommodate the resonant frequency drift caused by different application environments.Acoustic losses can be mitigated by positioning transducers at the peaks of interference patterns.Medium losses depend on the application environment, with high-frequency transducers offering better penetration but incurring higher losses.

In summary, this study developed a UET system model that spans from the input source to the load. By examining the energy transmission path, the study explored the link losses within the system. The model’s prediction results under various application environments were validated.

The proposed model was compared to related works. In this regard, applications are considered. The comparison factors are analyzed, such as transfer media materials, transmission distance, frequency range, and errors. The average error was calculated based on the comparison of the model with the measured results. The comparison is described in Table 4.

This model not only compares performance across different transmission media but also analyzes transmission distance continuously rather than at discrete points. Additionally, the model accounts for variations in frequency and impedance of the transducer under different application conditions. The model demonstrates an error margin of less than 3.2% for air and metal media. Although the error is higher for water, this discrepancy is attributed to the testing conditions. Using a larger water tank for testing could reduce wall interference, thereby decreasing the error.

## 5. Conclusions

The proposed ultrasonic energy transmission model provides a comprehensive explanation of the energy transmission process, accounting for several types of losses, including electrical, transducer, acoustic, and medium losses, across different environments. The model offers critical guidance on key factors such as impedance matching and acoustic matching, which are essential for optimizing the practical application of ultrasonic energy transmission.

Experimental results agree with the model’s predictions. The error analysis has shown that the maximum error between theoretical and experimental responses is 3.11% (air), 27.37% (water), and 1.76% (aluminum), which confirms its capability to simulate and guide the design of UET systems effectively. These validations demonstrate that the model can predict performance in different transmission media, helping optimize the efficiency and reliability of energy transfer.

However, the study’s scope was limited by certain experimental constraints, including the range of media and transducer conditions assessed. To further improve the model and explore its broader applications, future work will include testing under varying conditions, such as different atmospheric pressures, water depths, and metal types. These additional tests will improve the model’s accuracy and its applicability to more complex and diverse real-world environments.

## Figures and Tables

**Figure 1 sensors-24-06230-f001:**
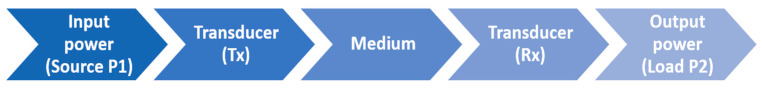
The UET system energy flow diagram.

**Figure 2 sensors-24-06230-f002:**
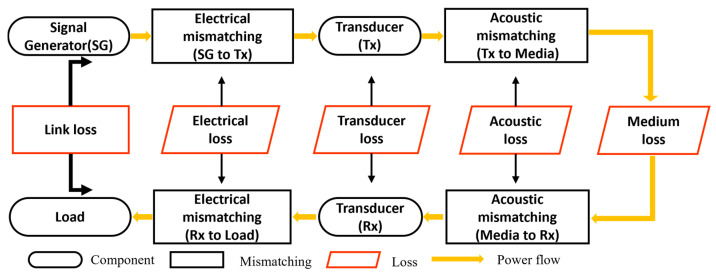
Flow diagram of transmission link loss. System loss mainly includes electrical loss, transducer loss, acoustic loss, and dielectric loss.

**Figure 3 sensors-24-06230-f003:**
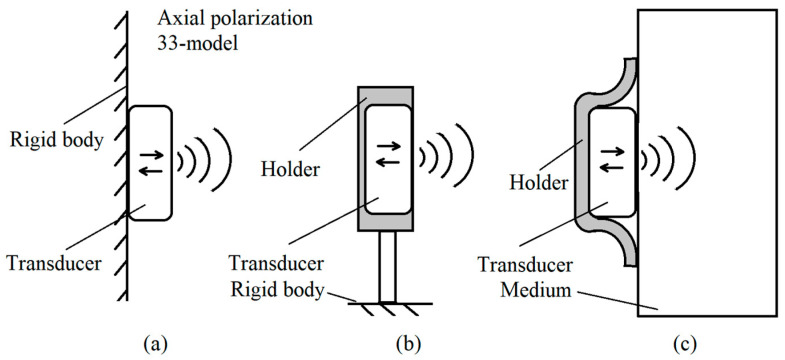
Schematic diagram of application of infinite planar piston model. This diagram examined three different configurations for fixing the ultrasonic transducer. The transducer’s vibrating surface moves along the axis, emitting ultrasonic waves in one direction: (**a**) Transducer fixed on a rigid body, (**b**) transducer fixed via holder on a rigid body, (**c**) transducer directly attached to the medium via holder.

**Figure 4 sensors-24-06230-f004:**
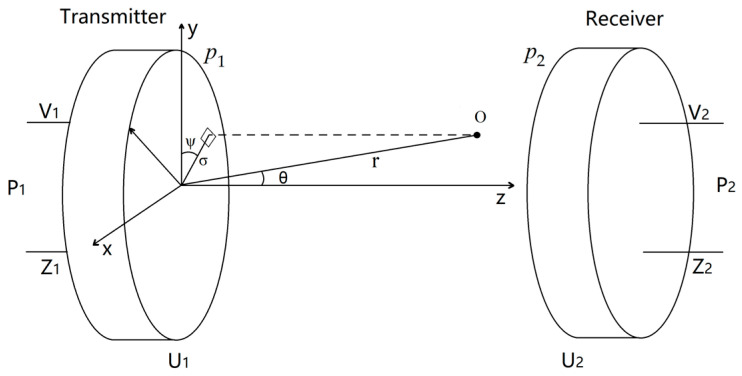
The transmitter with radius a and impedance Z_1_ emits ultrasonic waves outwards, driven by the input power P_1_ and voltage V_1_. The emitted pressure is *p_1,_* and the amplitude of surface vibration velocity is U_1_. The receiver with the same radius and impedance is Z_2_. The pressure at the receiver is *p*_2_, the amplitude of the surface vibration velocity is U_2_, the generated power is P_2_, and the voltage is V_2_. The piston lies in the x–y plane and vibrates vertically parallel to the z-axis. The distance between the observation point *O* and the center of the piston is *r*, and the angle between the e z-axis is *θ*.

**Figure 5 sensors-24-06230-f005:**
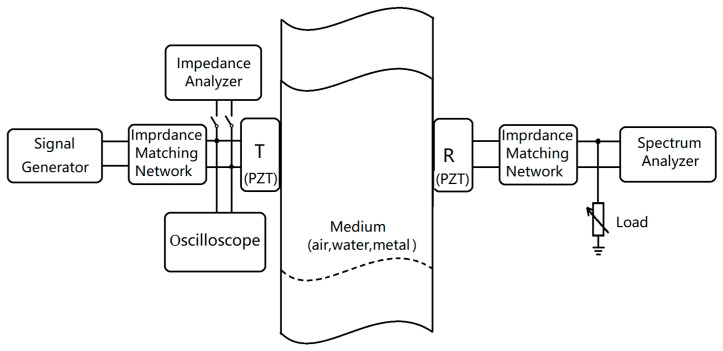
Illustration of a UET system including signal generator, impedance matching networks, transmit transducer (T), transmission medium (air, water, or metal), receive transducer (R), and spectrum analyzer (from left to right). An impedance analyzer and a spectrum analyzer are used to measure the input and output impedance of the T and R transducers, respectively. An oscilloscope is used to monitor the input voltage.

**Figure 6 sensors-24-06230-f006:**
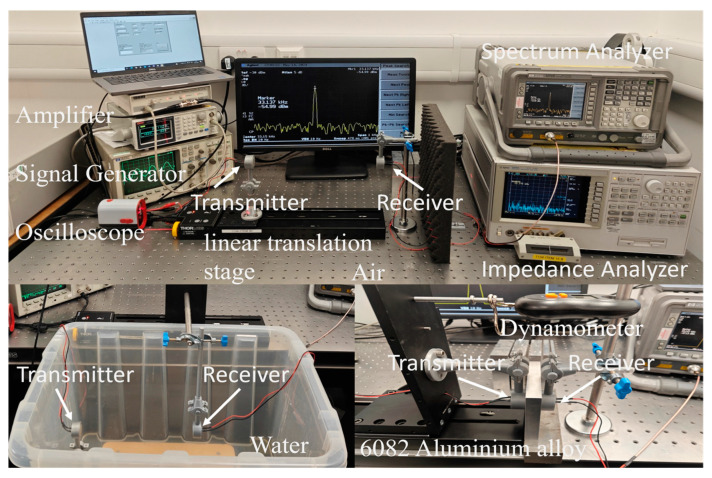
Experimental setups for verifying the transmission model for three different scenarios: air, underwater, and metal.

**Figure 7 sensors-24-06230-f007:**
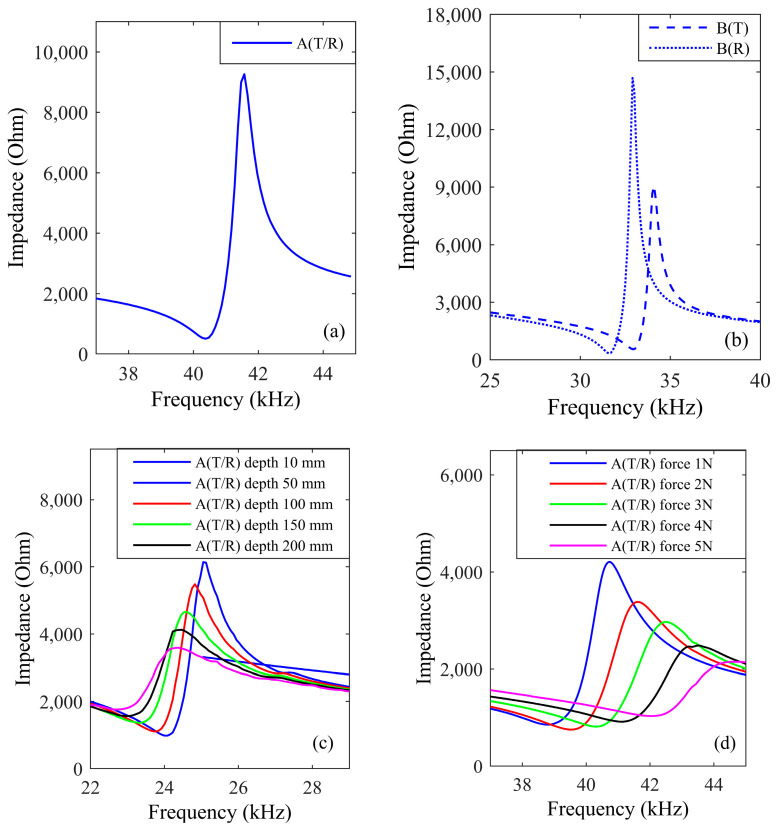
The impedance characteristics of transducers *A* and *B* for (**a**,**b**) air, (**c**,**e**) water, and (**d**,**f**) aluminum. The air test’s condition was set at the laboratory ambient atmospheric pressure and 25 °C. For water, the transducer was submerged at depths of 10 mm, 50 mm, 100 mm, 150 mm, and 200 mm. For aluminum, the pressure is applied at both ends of the transducer, with values of 1 N, 2 N, 3 N, 4 N, and 5 N.

**Figure 8 sensors-24-06230-f008:**
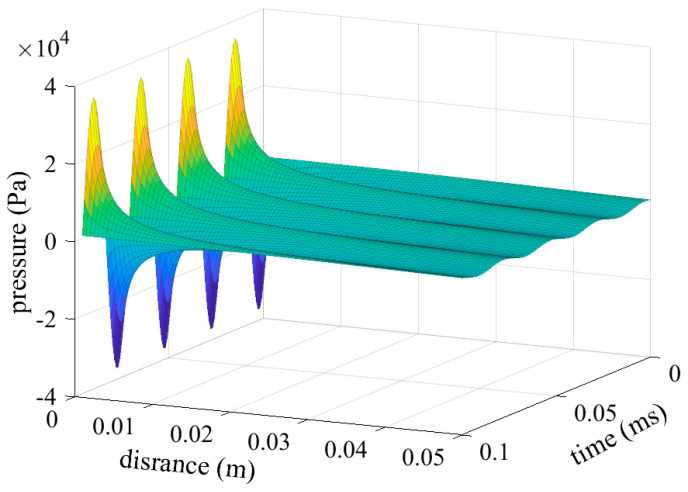
The relationship between pressure amplitude and distance and time in an ideal free space without reflection. The sound pressure amplitude decreases with distance due to medium losses. Additionally, since the driving source is a sine wave, the sound pressure oscillates over time as the frequency of the source.

**Figure 9 sensors-24-06230-f009:**
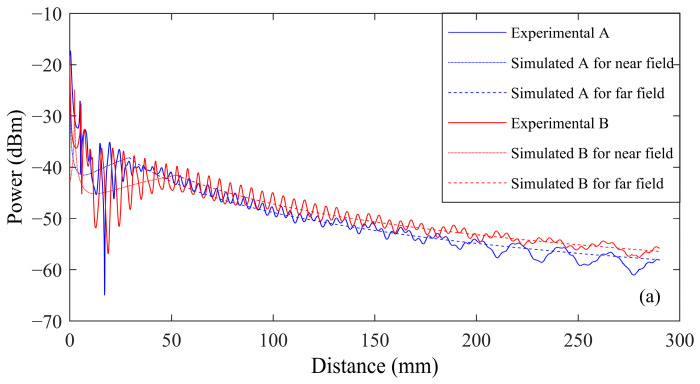
Transducer energy transmission test results for three transmission media: (**a**) air, (**b**) water, and (**c**) metal. In the setup, the transmitter is fixed at one end and driven by a 20 dBm source. Meanwhile, the receiver is fixed on the other end and connected to a spectrum analyzer. The results were recorded by sampling at different transmission distances.

**Table 1 sensors-24-06230-t001:** The parameters used in this study [31].

Material	Parameters	Unit	Value
Air	density	kg/m^3^	1
	sound velocity	m/s	343
	acoustic impedance	10^6^ × kg/m^2^ s	0.0034
Water	density	kg/m^3^	997
	sound velocity	m/s	1497
	acoustic impedance	10^6^ × kg/m^2^ s	1.49
Aluminum	density	kg/m^3^	2700
	sound velocity	m/s	6420
	acoustic impedance	10^6^ × kg/m^2^ s	17.33
Transducer A	frequency	kHz	40
	diameter	mm	18
	acoustic impedance	10^6^ × kg/m^2^ s	30.8
Transducer B	frequency	kHz	33
	diameter	mm	25
	acoustic impedance	10^6^ × kg/m^2^ s	30.8

**Table 2 sensors-24-06230-t002:** Link losses of different media.

Loss (dB)	Transducer	Air	Water (Depth = 150 mm)	Aluminum (Force = 5 N)
Electrical Loss	B (T)	5.19	11.49	5.65
	B (R)	15.54	10.27	18.86
	A (T/R)	4.88	9.21	4.37
Transducer Loss	B (T)	10.33	6.20	11.37
	B (R)	9.79	7.89	10.13
	A (T/R)	11.25	8.10	10.22
Acoustic loss	B (T)	33.55	5.35	0.35
	B (R)	33.55	5.35	0.35
	A (T/R)	33.55	5.35	0.35

Note: The driving source and load impedance were set to 50 Ω. Water tests were conducted at a depth of 150 mm, while a force of 0.5 N was applied in the aluminum tests.

**Table 3 sensors-24-06230-t003:** Error analysis for different media.

Medium	Transducer	Average Error
Air	A	2.54%
	B	3.11%
Water	A	27.37%
	B	19.68%
Aluminum	A	1.12%
	B	1.76%

**Table 4 sensors-24-06230-t004:** Comparison with related works.

Work	Medium	Distance	Frequency	Error
[15]	Stainless steel	74.8 mm	4 MHz	-
[16]	Alumina ceramic	50 mm	97–300 kHz	1.59–3.64%,
[17]	Aluminum	100 mm, 200 mm	738 kHz 1040 kHz	10.2–10.6%, 5.0–7.6%
	Steel	100 mm, 200 mm	738 kHz	6.1–6.9%
[18]	Aluminum	1.5 mm	0.353–1.96 MHz	1.14–2.67%
[19]	Aluminum	50 mm	656–978 kHz	21%
Present work	Air Water Aluminum	0–290 mm 0–290 mm 0–35 mm	19–42 kHz	2.54–3.11% 19.68–27.37% 1.12–1.76%

## Data Availability

All data supporting the findings of this study are contained within the article. Additional information related to this study is available from the corresponding author upon reasonable request.

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
