# Peer review of "A Universal Model for Ultrasonic Energy Transmission in Various Media"

_sensors, 2024, doi:10.3390/s24196230_

Round 1

Reviewer 1 Report

Comments and Suggestions for Authors

This paper presents a detailed model for ultrasonic energy transfer (UET) using a 33-mode piezoelectric transducer. The goal is to improve the powering of wireless sensors in difficult settings. UET, which utilizes ultrasound waves to transmit energy wirelessly over long distances, presents notable benefits and is attracting great research interests. Although there have been advancements in UET, current models mostly concentrate on assessing feasibility and doing numerical analysis, without adequately establishing a comprehensive connection between input and output power. The proposed model fills this need by integrating crucial elements such as resonant frequency, impedance matching, acoustic coupling, and boundary conditions, resulting in a more precise estimation of energy transfer efficiency. The model's performance is verified through a blend of numerical simulations and empirical tests carried out in diverse environments, encompassing air, metal, and underwater conditions.

Generally, the paper is well-written with solid experimental results. I personally think it can be accepted after minor corrections.

1. Please shorten the abstract and make it brief.

2. Figure 1 could be deleted for an academic paper.

3. Eq.1 is wired. Please use formal math.please check through the paper.

4. Eq.4 and 5 are classic equations. Can be cited.

5. Section 2 discussed a lot of common knowledge, which can be shortened.

6. Fig. 5 is blurry, please use clear figures.

7. Please make comments on the multiple reflections on the accuracy of the test.or maybe comments on the error analysis, if it applies. For example, the error analysis on fig. 10 and etc.

8. Transducer losses can be reduced by using wide-bandwidth transducers, please elaborate this point.

9. The biggest contribution, if i understand correctly, is to derive a detailed ultrasonic energy transfer (UET) to predict the efficiency.If so, contexts should be modified to further highlight how detailed“ the proposed model differ from the existing.

The last comment, the authors might cite more recent studies on impedance matching for electroacoustical transducers, like

  Digital non-Foster-inspired electronics for broadband impedance matching. Nat Commun 15, 4346 (2024). 

Comments on the Quality of English Language

can be improved.

Reviewer 2 Report

Comments and Suggestions for Authors

This study presents a comprehensive ultrasonic energy transfer (UET) model using a 33-mode piezoelectric transducer to enhance wireless sensor powering under challenging environments. However, the paper's contribution appears limited, and its structure is poorly organized, requiring reorganization. Several points must be addressed before the paper can be considered for publication in Sensors. The following revisions are recommended:

  1. The abstract must be revised to include the study's problem, methodology, and qualitative results.
  2. Significant results should be added to the abstract.
  3. The delivered power from UET is smaller than that obtained through capacitive and inductive coupling. This concept should be discussed, and the justification for using UET should be provided.
  4. In the introduction, the discussion of related works should be expanded to include further studies, such as that found at https://doi.org/10.3390/jlpea9030020
  5. The paper's contribution should be presented at the end of the introduction section.
  6. In lines 78-81, revise the sentence related to the paper's structure to reference "Sections" instead of "Chapters."
  7. Review Equation (1); it should be "P2 = P1 - Link loss."
  8. In lines 166-118, ensure the symbols defined in Equation (3), such as ZSG, ZRX, ZTX, and ZLoad, are consistent with those used in the Equation itself.
  9. All symbols in the equations should be clearly defined.
  10. Clearly label all parts of Figure 4.
  11. The properties presented in Table 1 should be supported with references for the values provided.
  12. Specific labels in Figure 6, such as "oscilloscope," do not correspond accurately to the devices shown in the diagram.
  13. The structure of the results and discussion section is disorganized and requires restructuring.
  14. The results and discussion sections need more in-depth discussion, especially regarding the results shown in Figures 8, 9, and 10.
  15. In Figure 10, clarify why Figure 10c (Aluminum alloy) is tested for shorter distances (0-30 mm) compared to Figures 10a and 10b, which are tested over a 0-300 mm range.
  16. Some references, such as [25, 26, 27, 28], are outdated; they should be updated.

Comments on the Quality of English Language

None

Reviewer 3 Report

Comments and Suggestions for Authors

This study introduces a comprehensive model for ultrasonic energy transfer (UET) utilizing a 33-mode piezoelectric transducer, aimed at advancing wireless sensor powering in challenging environments. The paper required major modification before considering for publication. The comments are as follows:

1. In the abstract, the authors did not briefly mention how the work has been conducted. In addition, no numerical results are provided to prove the effectiveness of the proposed energy transfer model for UET. Some quantative result should be presented in the abstract section in order to highlight the outcomes.

2. Introduction should divided into few parts for better understanding of the readers. My suggestion is to divide in four parts (i) Motivation and background (ii) literature review (iii) Gap of the research and Contribution (iv) paper organization.

3. The literature review part is limited to 5 references only. Many recent publications should be added and discussed to enhance the state-of-art. 

4. The contributions of this work should be improved and more clearly presented in bullet points.

5. Which type of WPT technique is applied for wireless transfer should be clearly mentioned. 

6. Check the arrow direction of the link loss in Figure 3.

7. As the authors are aiming to develop a model to show the relationship between input and output power, they have to propose an mathematical expression by providing information how different losses are incorporated in efficiency calculation. 

8. The authors should provide information on the coil design for WPT method and what is the distance between the Tx and Rx coils. 

9. A parameter table for the experimental table should be included. 

10. The authors have used many parameters in the manuscript such as in figure 8, A H1, A H2, B(T)1 etc. However, there are no descriptions provided for those parameters. They should include a list of abbreviations and notations in the appendix.

11. In Figure 8, legends are shown for 6 parameters. However, in the figure only 4 parameters are visible. It seemed Simulated A and B for far fields results are missing. Furthermore, what is the significance of these results is not clearly discusses. 

12. The authors claimed that the proposed model can provide a more accurate prediction of energy transfer efficiency. However, from the results there are no evidences of such claim has been found. The authors should extensively provide numerical results to prove this statement.

13. The most important thing missing in the manuscript is a comparative study between the proposed model and previous studies model (Discussed in literature review) based on numerical results. 

14. Conclusion should include findings of the study supported by numerical data obtained in the results. 

Comments on the Quality of English Language

Minor editing of English language required.

Round 2

Reviewer 2 Report

Comments and Suggestions for Authors

The authors have addressed all the reviewers' comments. Therefore, the paper can be accepted for publication in its current form.

Comments on the Quality of English Language

None

Reviewer 3 Report

Comments and Suggestions for Authors

The authors have addressed all the comments.